# Predictive Control and Regret Analysis of Non-Stationary MDP with Look-ahead Information

**Ziyi Zhang**                                                                    *ziyizhan@andrew.cmu.edu*
*Department of Electrical and Computer Engineering*
*Carnegie Mellon University*

**Yorie Nakahira**                                                                *yorie@cmu.edu*
*Department of Electrical and Computer Engineering*
*Carnegie Mellon University*

**Guannan Qu**                                                                    *gqu@andrew.cmu.edu*
*Department of Electrical and Computer Engineering*
*Carnegie Mellon University*

**Reviewed on OpenReview:** *https://openreview.net/forum?id=uObs1YwXjQ*

## Abstract

Policy design in non-stationary Markov Decision Processes (MDPs) is inherently challenging due to the complexities introduced by time-varying system transition and reward, which make it difficult for learners to determine the optimal actions for maximizing cumulative future rewards. Fortunately, in many practical applications, such as energy systems, look-ahead predictions are available, including forecasts for renewable energy generation and demand. In this paper, we leverage these look-ahead predictions and propose an algorithm designed to achieve low regret in non-stationary MDPs by incorporating such predictions. Our theoretical analysis demonstrates that, under certain assumptions, the regret decreases exponentially as the look-ahead window expands. When the system prediction is subject to error, the regret does not explode even if the prediction error grows sub-exponentially as a function of the prediction horizon. We validate our approach through simulations and confirm its efficacy in non-stationary environments.

## 1 Introduction

Policy design of non-stationary Markov Decision Processes (MDPs) has always been challenging due to the time-varying system dynamics and rewards, so the learner usually suffers from uncertainties of future rewards and transitions. Fortunately, exogenous predictions are available in many applications. For example, in energy systems, look-ahead information is available in the form of renewable generation forecasts and demand forecasts (Amin et al., 2019). It is intuitive to design an algorithm that controls the energy system by utilizing that information to concentrate energy usage in the time frame with the lowest energy price and lower the overall energy cost. To give another example, smart servers can make predictions of future internet traffic from historical data (Katris & Daskalaki, 2015). Given that the dispatcher tries to minimize the average waiting time of all tasks, if there is only light traffic, the average waiting time will be most reduced by only using the fastest server. If the dispatcher forecasts that there will be heavy traffic in the future, all servers should work to reduce the length of the queue.

However, although policy adaptation in a time-varying environment has been extensively studied (Auer et al., 2008; Richards et al., 2021; Zhang et al., 2024; Gajane et al., 2018), they do not typically take advantage of exogenous predictions. One branch of work focuses on adaptation and relies on periodic reset of the controller (Auer et al., 2008) or prior knowledge of the environment (Richards et al., 2021; Zhang et al., 2024). However, prior knowledge of the environment is usually difficult to obtain, and periodically resetting

the controller generates a linear regret. Another branch of work focuses on predicting the future based on past data. As past data can not accurately reflect the system in the future, they often need a previously specified (small) variation budget (Merlis, 2024; Lee et al., 2024; Gajane et al., 2018; Padakandla et al., 2020) to achieve sublinear regret. Overall, existing works demonstrate significant challenges in achieving sublinear regret under general non-stationary MDP on a single trajectory without assumptions on sublinear variation budget.

Most of the above algorithms do not utilize exogenous predictions widely available in applications. With the availability of predictions, it is natural to reason that if we could obtain accurate predictions of the entire future, we would easily obtain the optimal policy (thus zero regret). Furthermore, even if accurate predictions of the full future are not possible, it is reasonable to expect that these imperfect predictions can help the decision maker. Given these intuitions, we ask the question: with potentially imperfect prediction of transition kernel and reward function into the future, can we design an algorithm that leverages the prediction to obtain a sublinear regret with reasonable length and accuracy of the prediction?

**Contribution:** We propose an algorithm, Model Predictive Dynamical Programming (MPDP), that utilizes predictions of transitional probability and reward function to minimize the dynamic regret under the setting of a single trajectory. We utilize the look-ahead information and the *span* semi-norm to show that, under certain assumptions, the regret decays exponentially with the length of the prediction horizon when the prediction is error-less. Specifically, we show that MPDP achieves a regret of $O(T\gamma^{\lfloor k/J \rfloor}D)$, where $T$ is the time horizon, $k$ is the prediction horizon, $\gamma < 1$, and $J, D$ are constants determined by the properties of the MDP. Even when the system prediction is subject to error, we demonstrate that the regret does not explode if the growth rate of error is subexponential as a function of the prediction horizon. To the best of our knowledge, this paper is the first paper that explores the use of exogenous predictions in non-stationary MDP without prior knowledge and proposes an algorithm with decaying regret as the prediction horizon increases and the prediction error descreases.

The key technique underlying our result for sublinear regret is the contraction of the Bellman operator under the *span* semi-norm. We show that the value function of MPDP converges to the optimal value function exponentially in the *span* semi-norm under certain assumptions, which leads to the overall sublinear regret. Our result serves as the first step towards future applications of model predictive control in non-stationary RL with no prior information.

## 1.1 Related Works

**Non-stationary reinforcement learning** Many works have been done for reinforcement learning (RL) in non-stationary environment, in which a learner tries to maximize accumulated reward during its lifetime. For example, Merlis (2024); Lee et al. (2024) both use past data to predict future system dynamics via Temporal Difference (TD) learning. However, the standard TD methods are established on stationary MDP, so its applications on non-stationary MDP require a limited variation budge on a single trajectory (Li et al., 2019b; Lee et al., 2024; Wei & Luo, 2021; Chandak et al., 2020; Jali et al., 2025) or among a sequence of episodes (Feng et al., 2023; Moon & Hashemi, 2024; Zhao et al., 2022). For example, Jali et al. (2025) recently achieves a regret of $\tilde{O}(\sqrt{SA}\Delta_T^{\frac{1}{9}}T^{\frac{8}{9}})$, where $S$ is the size of state space, $A$ is the size of action space, $\Delta_T$ is the total variation budget, and $T$ is the time horizon. Other works reset the controller at the end of each episode (Gajane et al., 2018; Auer et al., 2008) and assume the ground-truth MDP stays the same within each episode. In this line of work, Merlis (2024) also utilizes prediction in policy design and achieves a regret of $O(H^3S^2A\sqrt{K}\ln(SATK))$, where $K$ is the number of total episodes. However, this work only allows for changes in either the reward function **or** the transitional probability while the other stays the same for all episodes, whereas in our setting, we allow for changes in both reward function and transitional probability and achieve an exponentially decaying regret. Another line of work actively detects the switch of MDPs by maintaining an estimate of system dynamics and resetting the controller when a switch is detected (Alami et al., 2023; Dahlin et al., 2023). More recently, a new branch of work uses exogenous predictions, but requires pre-trained optimal policies for each potential MDP the learner encounters, and designs the policy through a mixture of those optimal policies (Pourshamsaei & Nobakhti, 2024). Compared with those works, this paper focuses on a different setting of policy design on a single rollout, which does not

allow reset of system at the end of each episode for traditional non-stationary episodic RL. Furthermore, we utilize exogenous predictions for controlling a non-stationary MDP instead of generating those predictions from past data and do not require any prior knowledge about those MDPs.

**Model predictive control** Traditionally, model predictive control is a branch of control theory that, at each time step, calculates a predictive trajectory for the upcoming $k$ time steps and then implements the first control action from this trajectory (Lin et al., 2021). Some of these works seek to achieve guarantees such as static regret (Agarwal et al., 2019; Simchowitz & Foster, 2020), dynamic regret (Li et al., 2019a; Yu et al., 2020), or competitive ratio (Shi et al., 2020). From a theoretical perspective, extensive research has been conducted on the asymptotic properties of MPC, including stability and convergence, under broad assumptions about the system dynamics (Diehl et al., 2011; Angeli et al., 2012). Remarkably, Lin et al. (2021) shows that regret in a linear-time-varying system decays exponentially with the length of the prediction horizon. However, most of those works assume linear and deterministic system dynamics in continuous space in order to obtain a theoretical regret guarantee. In this paper, we focus on an MDP setting with finite state spaces and stochastic transition kernels, where the tools and algorithm design are very different.

## 2 Problem Formulation

We start by introducing the discrete non-stationary Markov Decision Process characterized by the tuple $(\mathcal{S}, \mathcal{A}, T, \{P_t\}_{t=0}^T, \{r_t\}_{t=0}^T)$. Here $\mathcal{S}, \mathcal{A}$ denote the discrete state space and action space, respectively. The set of functions $\{P_t(\cdot|s,a)\}_{(s,a)\in\mathcal{S}\times\mathcal{A}}$ is the collection of transition probability measures indexed by the state-action pair $(s,a)$ and time step $t$. The set of function $\{r_t(s,a)\}_{(s,a)\in\mathcal{S}\times\mathcal{A}}$ is the expected instantaneous reward, where $r_t(s,a)$ is the deterministic reward function taking value in $[0,1]$ incurred by taking action $a$ at state $s$ and time step $t$. We denote $T$ as the time horizon. Lastly, we use $\pi_t(\cdot|\cdot)$ to denote a decision policy at time $t$ that maps $s_t$ to $a_t$ and use $\boldsymbol{\pi}$ to denote the collection of $\{\pi_t\}_{t=0}^T$.

The learner does not have access to the transition probability $\{P_t\}_t$ and reward functions $\{r_t\}_t$. Instead, we consider a setting in which the learner can predict the future and obtain estimates of future transitional probabilities $\hat{P}$ and rewards $\hat{r}$ for $k$ time steps. More specifically, at any time $t$, the learner has reward and transitional probability estimation $\hat{r}_{t+\ell|t}, \hat{P}_{t+\ell|t}$ for any $\ell \in \{1, \ldots, k\}$. The error of the predictions are characterized in the below definition.

**Assumption 1** (prediction error)**.** The reward estimation $\hat{r}(s,a)$ and transitional probability estimation $\hat{P}$ has the error bound

$$|\hat{r}_{t+\ell|t}(s,a) - r_{t+\ell}(s,a)| < \epsilon_\ell,$$
$$\left\|\hat{P}_{t+\ell|t}(\cdot|s,a) - P_{t+\ell}(\cdot|s,a)\right\|_{\mathrm{TV}} < \delta_\ell,$$

for all $s, a, t$, where $\|\cdot\|_{\mathrm{TV}}$ denotes the total variation norm.

The fact that the prediction error is a function of the prediction distance $\ell$ is intuitive, as system forecasts are often more accurate in the near future than in the distant future.

The learner implements an algorithm ALG that at time $t$, observes $s_t$ and is given the prediction $\hat{r}_{t+\ell|t}, \hat{P}_{t+\ell|t}$ for any $\ell \in \{1, \ldots, k\}$. It generates a policy at step $t$, denoted as $\pi_t$, based on available information. Given the algorithm ALG, its value function is defined by

$$V_t^{\mathrm{ALG}}(s) = \sum_{i=t}^T \mathbb{E}^{\mathrm{ALG}}\left[r_i(s_i, a_i)|s_t = s\right]. \tag{1}$$

such that $s_{i+1} \sim P_i(\cdot|s_i, a_i)$ for all $i$, where $a_i$ is generated by $\pi_i$.

We also define the offline optimal, which is the optimal value had the learner known all the future transitions and rewards precisely. Formally, the optimal value function is defined by

$$V_t^*(s) = \max_{\pi=\{\pi_i\}} \mathbb{E} \sum_{i=t}^T [r_i(s_i, a_i)|s_t = s],$$

where in the expectation, $a_i \sim \pi_i(s_i)$.

The learner's objective is to design a policy $\pi$ that utilizes the dynamics forecast to maximize the cumulative reward and minimize the dynamic regret

$$\mathcal{R}(\text{ALG}) := V_0^*(s_0) - V_0^{\text{ALG}}(s_0), \tag{2}$$

for some fixed initial state $s_0$.

*Example* 1. A server needs to allocate resources for tasks and minimize the average wait time for each job. We consider the setting that there are $n$ servers, each has a service rate $\mu_i$, and a single queue with a time-varying arrival rate $\lambda_t$. The system's state space is the length of the queue and whether each server is idle or busy, and its action space is the decision of sending a job in the queue to an available server or not doing anything. The transitional probability is determined as follows. At each time, a job arrives at the queue with probability $\lambda_t/(\lambda_t + \sum_i \mu_i)$. Then, a dispatcher decides whether to send a job in the queue to one of the idle servers or wait till the next time step. Also, the busy servers will complete the job and become idle with probability $(\sum_i \mu_i)/(\lambda_t + \sum_i \mu_i)$. For a detailed explanation of the setup, see Jali et al. (2024). In this example, the job arrival rate $\lambda_t$ varies in time, and having a prediction of the future arrival rate can help the dispatcher in making the job assignment decision. Existing methods for future Internet traffic forecasting (Katris & Daskalaki, 2015) can greatly help with the process and reduce the average wait time.

*Example* 2. Renewable energy generation is heavily influenced by environmental factors, and its prediction is also subject to error with different prediction horizons. We consider a discrete-time model for electric vehicle (EV) charging. A list of EV arrives at the charging station from time step 0 to $T$. The $i$-th EV arrives at the station at time $a_i$, if the station is already full, it would leave the station without charging. If there is an unused charging stand, it sets a departure time $d_i$ and an energy demand $e_i$. Moreover, each charging stand has a charging rate capacity $\mu$, and the station has an overall charging capacity $C$. The station needs to charge all electric vehicles that arrive by setting a charging rate $r_{i,t}$ for each vehicle $i$ at each time step $t$. The energy demand of each vehicle has to be satisfied, i.e. $\sum_{t=a_i}^{d_i} r_{i,t} = e_i$. Moreover, the charging rate has to be below the threshold of each charging stand and the station, i.e. for every $t$, $r_{i,t} \in [0, \mu], \sum_{i=1}^{3} r_{i,t} < C$. For a detailed explanation of the setup, see Chen et al. (2022). Moreover, the energy price fluctuates with energy supply and demand. Since the energy price fluctuates with time $t$, the station wants to minimize the total cost of energy while still fulfilling the energy demand of all vehicles before their departures. In this setup, the state space is the energy demand of each EV, the energy price, and the time frame in which those demands have to be met. The action space is the decision of whether to supply the required amount of energy now or to wait for a later time. As both EV parking demands and energy price can be forecasted (Provoost et al., 2020; McHugh et al., 2022; Kapoor et al., 2025), learners can utilize this information to adjust their policies.

## 3 Preliminary

Before we introduce the algorithm, we introduce a few concepts that play a key role in the decay of regret.

### 3.1 Span Semi-norm and Bellman Operator

First, we introduce the *span* semi-norm.

**Definition 1.** For any vector $v \in \mathbb{R}^d$, define the *span* seminorm of $v$, denoted as $\text{sp}(v)$ by

$$\text{sp}(v) := \max_i v(i) - \min_i v(i),$$

where $v(i)$ is the $i$'th entry of $v$.

The properties of *span* semi-norm is briefly described in Proposition A.1 in Appendix A. More description can be found in Puterman (1994). We use *span* semi-norm to quantify our approximation of the $Q$ functions. Since we use $\arg\max$ on the $Q$ function to determine which action the learner should take, shifting the entire $Q$ function by a constant does not affect the policy. Since *span* semi-norm has this property (see Proposition A.1 in Appendix A), we use *span* semi-norm to quantify the deviation of our estimated $Q$ function from the ground-truth $Q$ function.

We also introduce the Bellman operator in the vector format:

$$L_t v(s) = \max_{a \in \mathcal{A}} \left\{ r_t(s, a) + \sum_{s' \in \mathcal{S}} P_t(s'|s, a) v(s') \right\}, \tag{3}$$

where $v \in \mathbb{R}^{|\mathcal{S}|}$ is the vector of value function for all states. Similarly, define

$$L_t^{\pi} v(s) = \mathbb{E}_{a \sim \pi(s)} \left[ r_t(s, a) + \sum_{s' \in \mathcal{S}} P_t(s'|s, a) v(s') \right]. \tag{4}$$

Furthermore, we denote the time-varying nested Bellman operator as follows:

$$L_{t:t'} := L_t \circ \cdots \circ L_{t'} \tag{5}$$

$$L_{t:t'}^{\pi_t:\pi_{t'}} := L_t^{\pi_t} \circ \cdots \circ L_{t'}^{\pi_{t'}}. \tag{6}$$

where $f \circ g := f(g(\cdot))$ is the nested operator. Similarly, we use $P_t^{\pi}$ and $P_t^a$ to denote the transitional probability matrix when using policy $\pi$ or taking action $a$ at time $t$, respectively. We use $P_{t:t'}^{\pi}$ and $P_{t:t'}^{a_t:a_{t'}}$ to denote the nested product of the transitional probabilities.

We now introduce the $J$-stage contraction.

**Definition 2** ($J$-stage contraction). We say a sequence of operators $\{F_t\}_{t \in \{1,\ldots,T\}} : \mathcal{S} \to \mathcal{S}$ is a $J$-stage span contraction if exists $\gamma \in (0, 1)$, such that

$$\mathrm{sp}(F_t \circ \cdots \circ F_{t+J} u - F_t \circ \cdots \circ F_{t+J} v) < \gamma \, \mathrm{sp}(u - v),$$

for all $u, v \in \mathbb{R}^{|\mathcal{S}|}, t \in \{1, \ldots, T - J\}$.

**Assumption 2.** There exists $J \in \mathbb{Z}^+$ such that for the optimal policy $\boldsymbol{\pi}^* = \{\pi_h^*\}_{h=0}^T$ and any other policy $\boldsymbol{\pi} = \{\pi_h\}_{h=0}^T$ at time step $t \in \{0, \ldots, T - J\}$,

$$\eta(\boldsymbol{\pi}, \boldsymbol{\pi}^*) = \min_{s_1, s_2 \in \mathcal{S}} \sum_{j \in \mathcal{S}} \min \left\{ P_t^{\pi_t} \circ \cdots \circ P_{t+J}^{\pi_{t+J}}(j|s_1), P_t^{\pi_t^*} \circ \cdots \circ P_{t+J}^{\pi_{t+J}^*}(j|s_2) \right\} > 0.$$

The above assumption is similar to the uniform ergodicity assumption in Yu & Mannor (2009); Li et al. (2019b). To see this, we note that Assumption III.1 of Yu & Mannor (2009) also requires ergodicity among states in a non-stationary environment under any pairs of policies. This assumption implies that the effect of any mistake decays exponentially with the number of passing time steps. In episodic non-stationary RL (e.g., Moon & Hashemi (2024); Zhao et al. (2022)), this assumption is automatically satisfied as the environment resets at the end of each episode, so the learner can recover any mistake in future episodes. Further, we point out that if Assumption 2 is not satisfied, it would imply that there exist certain situations within the MDP such that, for any $k$ and $T$, there exists an non-stationary MDP that violates Assumption 2, such that any algorithm would have expected regret of $O(T)$. We give a counter-example of such non-stationary MDP in Counter-example F.1.

In the next proposition, we introduce how Assumption 2 establishes contraction, which we later use to show the decay of regret.

**Proposition 3.1.** *For any non-stationary MDP satisfying Assumption 2, $L_t$ defined in equation 3 is a $J$-stage contraction operator with contraction coefficient*

$$\gamma = 1 - \min_{\boldsymbol{\pi}} \eta(\boldsymbol{\pi}, \boldsymbol{\pi}^*).$$

The proof of Proposition 3.1 is deferred to Appendix A.

### 3.2 Diameter

The *diameter* of MDP is commonly used in reinforcement learning (Gajane et al., 2018; Wu et al., 2022). We extend the definition to non-stationary MDP.

**Definition 3.** Given a non-stationary MDP, time $t$, state $s$, policy $\boldsymbol{\pi}$, and family of sets $\{S^{(i)} \subset \mathcal{S}\}_i$, define

$$d_t(s, \boldsymbol{\pi}, \{S^{(i)}\}_i) = \inf\{\tau : \tau > 0, s_{t+\tau} \in S^{(t+\tau)} | s_t = s\},$$

where $\{s_{t+\tau}\}_{\tau=0}^{\infty}$ is generated by policy $\boldsymbol{\pi}$, and let $\mathcal{T}(\{S^{(i)}\}_i | \boldsymbol{\pi}, s)$ denote the minimal travel time from $s$ to the family of sets $\{S^{(i)}\}_i$ starting at any $t$, i.e.

$$\mathcal{T}(\{S^{(i)}\}_i | \boldsymbol{\pi}, s) = \sup_t \left\{ \mathbb{E}[d_t(s, \boldsymbol{\pi}, \{S^{(i)}\}_i)] \right\}. \tag{7}$$

Let $S_i^* := \arg\max_s V_i^*(s)$ denote the set of states with the maximal value function at time step $i$. Define the *diameter* of a non-stationary MDP as

$$D = \max_s \min_{\boldsymbol{\pi}} \left\{ \mathcal{T}(\{S_i^*\}_i | \boldsymbol{\pi}, s) \right\}. \tag{8}$$

In this paper, the time horizon $T$ is finite. Therefore, when $i > T$, we have $S_i^* = \mathcal{S}$. As the agent will arrive at $\{S_i^*\}_i$ immediately after the $T$-th time step, we always have $D < T + 1$ for any finite horizon MDP.

**Proposition 3.2.** *For a stationary MDP, The diameter as defined in Definition 3 is upper bounded by the conventional diameter definition for stationary MDP as in Wu et al. (2022).*

The proof of Proposition 3.2 is straightforward, as the $\{s_{t+i}^*\}_i$ is constant for all $t, i$ in time-invariant MDP.

The diameter of a non-stationary MDP upper bounds the span semi-norm of the optimal value function, as we present in the following Proposition:

**Proposition 3.3.** *For any non-stationary MDP with diameter $D$ defined in Definition 3, $\mathrm{sp}(V_t^*) < D$ for all $t$.*

The proof of Proposition 3.3 is deferred to Appendix B.

## 4 Main Results

### 4.1 Algorithm Design

On a high level, at each time step $t$, the proposed algorithm Model Predictive Dynamic Programming (MPDP) works by conducting a dynamic programming style planning for the next $k$ steps, and takes the first action. More precisely, we define the Bellman operator $\hat{L}$ on system dynamics forecast as follows:

$$\hat{L}_{t+\ell|t} v(s) = \max_{\hat{a} \in \mathcal{A}} \left\{ \hat{r}_{t+\ell|t}(s, \hat{a}) + \sum_{s' \in \mathcal{S}} \hat{P}_{t+\ell|t}(s'|s, \hat{a}) v(s') \right\}, \tag{9}$$

where the learner optimize on the forecast of reward and transitional probability, instead of the ground-truth as in equation 3. The learner picks action $a$ such that

$$\begin{aligned}
a_t &= \arg\max_{a \in \mathcal{A}} \hat{r}(s_t, a) + \max_{a_{t+1}} \mathbb{E}[\hat{r}_{t+1}(s_{t+1}, a_{t+1}) + \max_{a_{t+2}} \mathbb{E}[\cdots + \max_{a_{t+k}} \hat{r}_{t+k}(s_{t+k, a_{t+k}})]] \\
&= \arg\max_{a \in \mathcal{A}} \hat{L}_{t|t} \circ \cdots \circ \hat{L}_{t+k|t} W_0,
\end{aligned} \tag{10}$$

where $W_0$ denotes the zero constant vector. Intuitively, the learner undergoes a dynamic programming process for the future $k$ steps based on the reward and transitional probability forecasts, and takes the first action of the dynamic programming. Then, the learner obtains a new forecast and repeats the process.

---

**Algorithm 1** Model predictive dynamical programming (MPDP)

---

1: Select $v^{(0)} \in \mathbb{R}^n$, specify $\epsilon > 0$, and set $S = 0$.
2: **for** $t = 0, 1, 2, \ldots, T$ **do**
3:     Forcast $\hat{P}_t, \ldots, \hat{P}_{t+k}, \hat{r}_t, \ldots, \hat{r}_{t+k}$
4:     Select $a_t$ according to equation 10.
5:     $s_{t+1} \sim P_t(\cdot | s_t, a_t)$.
6: **end for**

---

The algorithm is simple and intuitive. In line 3 of Algorithm 1, we forecast the system dynamics of the future $k$ steps. In line 4, we pick the first action that maximizes the reward in the future $k$ steps. The algorithm design is inspired by model predictive control (García et al., 1989), and we try to optimize the performance of the learner within the prediction horizon and decide the action by a dynamic programming style algorithm.

### 4.2 Regret Guarantee

In this section, we introduce our bound on regret as defined in equation 2.

**Theorem 4.1.** *For any non-stationary MDP satisfying Assumption 2, Algorithm 1 achieves a regret of*

$$\mathcal{R}(\text{MPDP}) \leq T\gamma^{\lfloor k/J \rfloor}D + 2T\epsilon_0 + 2T\delta_0 D + 2T \sum_{i=0}^{\lceil k/J \rceil - 1} \gamma^i \left( \sum_{j=1}^{J} \epsilon_{iJ+j} + \sum_{j=1}^{J} \delta_{iJ+j}D \right)$$
$$+ 2T\gamma^{\lfloor k/J \rfloor} \left( \sum_{j=1}^{k\%J} \epsilon_{\lfloor k/J \rfloor J + j} + \sum_{j=1}^{k\%J} \delta_{\lfloor k/J \rfloor J + j}D \right),$$

*where $k\%J := k - \lfloor k/J \rfloor \cdot J$.*

In the error-free setting (the prediction does not have error), Theorem 4.1 simplifies to the following corollary.

**Corollary 4.2.** *If the system dynamics forecast is exact for the future $k$ steps, then*

$$\mathcal{R}(\text{MPDP}) \leq T\gamma^{\lfloor k/J \rfloor}D.$$

We observe that, in the error-free case (Corollary 4.2), the regret depends linearly on the time horizon $T$ and diameter $D$ and decays exponentially with the prediction horizon $k$ when Assumption 2 is satisfied. In particular, Algorithm 1 with a log-prediction horizon $k = O(\log T)$ will obtain a regret sublinear in $T$. This means that predictions, even if a short horizon, are powerful in the sense that it leads to sublinear regret without any assumptions on the variation budget.

In the setting of inaccurate prediction (Theorem 4.1), we observe that regret grows linearly with the prediction error $\epsilon_\ell, \delta_\ell$. It is important to note that the sensitivity to $\epsilon_\ell, \delta_\ell$ decays exponentially in $\ell$, meaning that the regret is more sensitive to the prediction error of the near horizon than the long horizon. Therefore, even if the prediction error increases as $\ell$ increases, as long as the increase is subexponential, using predictions in the far future with potentially large errors still has a positive impact on the overall performance.

Lastly, our results also indicate a tradeoff between the error induced by the inaccurate predictions and the additional information it provides. For predictions further into the future, while it may contain valuable information, the potentially large error can also lead to a worse regret. We note that the learner can solve an optimization problem to determine the best $k$ to maximize on the exponential decay property and avoid the large error caused by forecasting too far into the future.

We briefly outline the steps of the proof of Theorem 4.1 in Section 5. The full proof is deferred to Appendix E.

# 5 Proof Outline

We split the proof outline into three separate steps. In the first step, we show that the estimated value function converges to the ground truth value function in the span semi-norm with increasing prediction horizon. In the second step, we bound the error of the estimated $Q$ function using the step-wise error bound of the Bellman operator. With the error of $Q$ function bounded, if the learner makes a mistake using the estimated $Q$ function, the maximal loss caused by that mistake can be bounded. Therefore, we can bound the maximal regret in the last step.

**Step 1: One step error bound.** As Algorithm 1 takes a greedy approach to optimize the reward within the prediction horizon of length $k$, we first need to approximate the optimal value function $V_t^*$ within the $k$ steps.

Let $\hat{P}_{t+\ell|t}$ denote the predicted transition probability for $(t+\ell)$-th step at time step $t$, and let $\hat{r}_t$ denote the predicted reward function. Let $\hat{\psi}_t^k$ denote the vector of the maximal expected reward at time $t$ for the next $k$ steps,

$$\hat{\psi}_t^\ell(s) = \begin{cases} 0, & t+\ell > T \text{ or } \ell > k, \\ \max_a \left\{ \hat{r}_{t+\ell|t}(s,a) + \mathbb{E}_{s' \sim \hat{P}_{t+\ell|t}(\cdot|s,a)} \hat{\psi}_t^{\ell+1}(s') \right\} & , \quad t+\ell \leq T, \ell \leq k. \end{cases} \tag{11}$$

Similarly, let $\tilde{\psi}_t^k$ denote the vector of the maximal expected reward with completely accurate forecasts.

$$\tilde{\psi}_t^\ell(s) = \begin{cases} 0, & t+\ell > T \text{ or } \ell > k, \\ \max_a \left\{ r_{t+\ell}(s,a) + \mathbb{E}_{s' \sim P_{t+\ell}(\cdot|s,a)} \tilde{\psi}_t^{\ell+1}(s') \right\} & , \quad t+\ell \leq T, \ell \leq k. \end{cases} \tag{12}$$

Since the forecast is different from the true system dynamics, we need to bound the difference between $\hat{\psi}_t^\ell$ and $\tilde{\psi}_t^\ell$ step-wise. First, we make a simple observation.

**Lemma 5.1.** *For a zero constant vector $W_0$, $\tilde{\psi}_t^0 = L_t \circ \cdots \circ L_{t'} W_0$, where $t' = \min\{T, t+k\}$. Similarly, $\hat{\psi}_t^0 = \hat{L}_t \circ \cdots \circ \hat{L}_{t'} W_0$.*

*Proof:* We proceed by induction. If $k = 0$, the equality is trivial. The induction step directly follows from equation 3 and equation 12.

$\square$

The above lemma implies that, when $t + k \geq T$, we have $\tilde{\psi}_t^0(s) = V_t^*(s)$ for all $s$. Furthermore, we point out that $V_t^* = L_t \circ \cdots \circ L_T W_0 = L_t \circ \ldots L_{t'} V_{t'+1}^*$. Correspondingly, we need to bound the error generated by each layer of the Bellman operators.

**Lemma 5.2.** *For any $\tilde{V}, \hat{V} \in \mathbb{R}^{|\mathcal{S}|}$ such that $\mathrm{sp}(\tilde{V}) < D, \mathrm{sp}(\hat{V}) < D$ and $\mathrm{sp}(\tilde{V} - \hat{V}) \leq b$, we have*

$$\mathrm{sp}\left(L_{t+\ell}\tilde{V} - \hat{L}_{t+\ell|t}\hat{V}\right) \leq b + 2\epsilon_\ell + 2\delta_\ell D.$$

The proof of the above lemma is left to Appendix C.

**Step 2: Bounding the error of $Q$ function.** We define the optimal $Q$ function as follows

$$Q_t^*(s,a) = r_t(s,a) + \mathbb{E}[V_{t+1}^*(s_{t+1})|s_t = s, a_t = a], \tag{13}$$

Since we use $\hat{\psi}$ and $\tilde{\psi}$ to estimate the value function $V_t^*$ of each time step, we can construct estimates of $Q_t^*$ as follows:

$$\tilde{\Psi}_t(s,a) := r_t(s,a) + \mathbb{E}_{s' \sim \mathbb{P}_t(\cdot|s,a)}[\tilde{\psi}_{t+1}^0(s')], \tag{14}$$

$$\hat{\Psi}_t(s,a) := \hat{r}_t(s,a) + \mathbb{E}_{s' \sim \hat{\mathbb{P}}_{t|t}(\cdot|s,a)}[\hat{\psi}_{t+1}^0(s')]. \tag{15}$$

In order to bound the error between $\hat{\Psi}_t$ and $Q_t^*$, we need to bound two pairs of the difference: the difference between $\tilde{\Psi}_t$ and $Q_t^*$, and the difference between $\tilde{\Psi}_t$ and $\hat{\Psi}_t$. The details of those steps are deffered to

Lemma D.3 and Lemma D.4 in Appendix D, respectively. With an error bound of our approximated $Q$ function, we can upper bound the loss of reward by each mistake made by the algorithm.

**Corollary 5.3.** *Let a be the action picked by Algorithm 1 with prediction error as defined in Assumption 1 at the t-th time step at state s, and let $a^*$ be the optimal action at time t, then,*

$$Q_t^*(s,a^*) - Q_t^*(s,a) \leq \gamma^{\lfloor k/J \rfloor} \operatorname{sp}\left(V_{t+k+1}^*\right) + 2\epsilon_0 + 2\delta_0 D + 2 \sum_{i=0}^{\lfloor k/J \rfloor - 1} \gamma^i \left( \sum_{j=1}^{J} \epsilon_{iJ+j} + \sum_{j=1}^{J} \delta_{iJ+j} D \right)$$
$$+ 2\gamma^{\lfloor k/J \rfloor} \left( \sum_{j=1}^{k\%J} \epsilon_{\lfloor k/J \rfloor \cdot J + j} + \sum_{j=1}^{k\%J} \delta_{\lfloor k/J \rfloor \cdot J + j} D \right).$$

The proof of Corollary 5.3 is deferred to Appendix D.

**Step 3: Bounding regret.** Since the error of $Q$ function is bounded at every step, we can upper bound the regret $R$ by bounding the telescoping sum of $\mathbb{E}[\sum_t (Q_t^*(s_t, a_t^*) - Q_t^*(s_t, a_t))]$. By bounding the error at each step, we obtain Theorem 4.1.

# 6 Simulation

## 6.1 Queueing system

In the first simulation, we simulate a queueing system based on the setup provided in Example 1. Specifically, we consider a representative example of 3 servers whose service rates $\{\mu_i\}_{i=1,2,3}$ are 100, 10, 1, respectively, with time horizon $T = 100$ and varying load $\lambda_t$ fluctuating from 10 to 100.

In the first part of this simulation, we compare regrets of different lengths of the prediction horizon $k$ and the Fast Available Server (FAS) (Lin & Kumar, 1984), ratio-of-service-rate-thresholds (RSRT) algorithm (Özkan & Kharoufeh, 2014), and Non-Stationary Natural Actor-Critic (NS-NAC) (Jali et al., 2025). FAS is a popular algorithm frequently used in practice, which sends any available job immediately to the fastest available server. RSRT is a threshold policy where a job is routed to the fastest among the available servers only if the queue length exceeds a predetermined threshold (Özkan & Kharoufeh, 2014). It has been proven to be the optimal policy in the two-server setting (Özkan & Kharoufeh, 2014). Notably, typical time-varying RL algorithm, such as NS-NAC, would not work in this example without a pretrained initial policy, as there are infinitely many states in the queuing problem, and the policy would keep exploring new states with longer queue lengths. For a proper comparison, we set the initial policy similar to RSRT with a small exploration probability for every action. We then compute regret where the optimal policy has full knowledge of the transitional probability. For each $k \in \{1, \ldots, 15\}$, we run 20 trials and record the average regret for each $k$ value. The agent has access to the predicted arrival rate of jobs with some Gaussian additive prediction error $\hat{\lambda}_t := \lambda_t + \mathcal{N}(0, \sigma)$ with $\sigma \in \{0, 1, 2\}$. The optimal policy we compute our regret from is the policy that is computed knowing all future transitional probability and reward functions. The arrival of jobs fluctuates periodically with a sin function with magnitude 110 to simulate periodical change in demand. The queue length of FAS is consistently the longest throughout the time horizon, and RSRT has a queue length similar to that of MPDP with $k = 8$. MPDP with $k = 12$ has the shortest queue length throughout most of the time steps. As shown in the rest of Figure 1, the proposed algorithm outperforms both benchmarks with $k \geq 8$ under noise-free setting and also outperforms both benchmarks with $k \geq 10$ with prediction error. Specifically, we see a decay in log-scale of regret. However, as we have $J$-stage contraction, the regret does not necessarily decrease monotonically with every increase in $k$.

In the second part of the simulation, we fix $k = 10$ and examine the relationship between the magnitude of the prediction error and regret more closely. Although the learner can still forecast the system dynamics in the future, the predicted arrival rate of jobs $\hat{\lambda}_t := \lambda_t + \mathcal{N}(0, \sigma_{t,\ell})$. We first hold $\sigma_{t,\ell}$ to be constant of $t, \ell$ as in the first part. Therefore, the variance of the prediction error does not increase with the distance of the forecast into the future. As shown in Figure 2a, the regret initially remained minimal and increased linearly after variance reaches 8, as shown in Theorem 4.1.

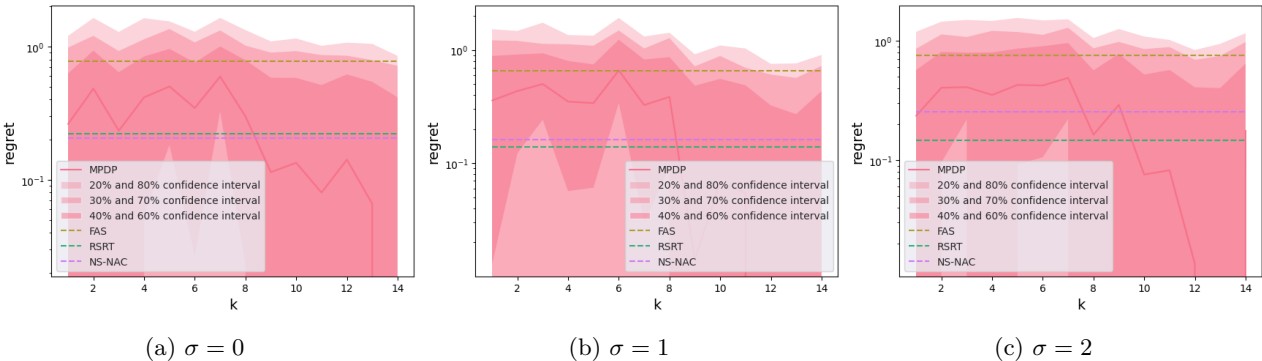

(a) $\sigma = 0$         (b) $\sigma = 1$         (c) $\sigma = 2$

Figure 1: Figure 1a, Figure 1b, and Figure 1c show the regret of MPDP under different additive prediction error $\mathcal{N}(0, \sigma)$. The red solid line shows the mean of the regret, and the shaded area shows the confidence interval.

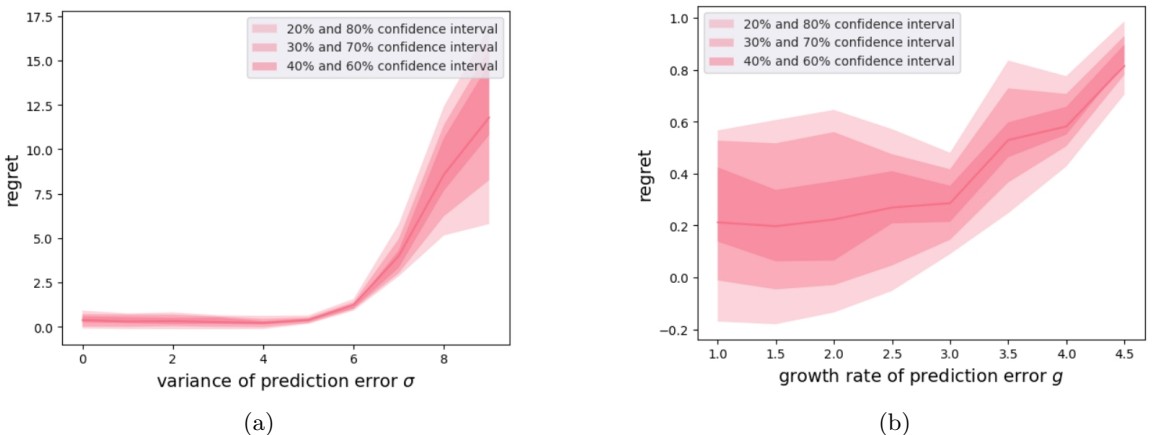

(a)                    (b)

Figure 2: Figure 2a shows the average regret remains almost constant with variance of prediction error below 5 and starts to grow afterward. Figure 2b shows the regret slowly increases with the growth rate of variance of the prediction error with respect to the prediction horizon.

Most practical forecast are usually more accurate for closer future than distant future. In applications like wind power generation, it has been shown that the accuracy of forecasts decreases at a linear rate with respect to the distant in the future (Qu et al., 2013). Therefore, in the third simulation, we fix $k = 10$ and the initial variance of the prediction error to be 1 and examine the relationship between regret and growth rate of variance with respect to the prediction horizon $g$. More specifically, for each forecast, we fix the variance $\sigma_{t,0} = 1$, and $\sigma_{t,\ell} = g * \ell$. We are most interested in the case where $g > 1$, as the variance tends to increase with respect to the distance to the future predicted by the forecast. As shown in Figure 2b, the regret does increase with growth rate $g$, but the increase is relatively slow. Even for the case $g = 4$, which indicates the variance increases by 4 times for every time step, thus reaching 40 times of the initial error at the end of the prediction horizon, the average regret merely increased 2 times. Indeed, by the expression in Theorem 4.1, the regret would never explode if the growth rate of variance is sub-exponential.

Lastly, we introduce a simplified version of the queueing problem in Example 1. While keeping all other factors identical, we fix $\lambda_t \in \{30, 130\}$ and switch $\lambda_t$ every 50 steps. By fixing $\lambda$ to only two potential values, we simplify the setting where a traditional RL algorithm with context detection might have an advantage. To incorporate more changes in MDP, we also extend the time horizon to $T = 300$.

The result of the simulation is shown in Figure 3. We introduce RL-CD proposed in da Silva et al. (2006) as a new benchmark. For a proper comparison, we also set the initial policy to be similar to RSRT, with a small

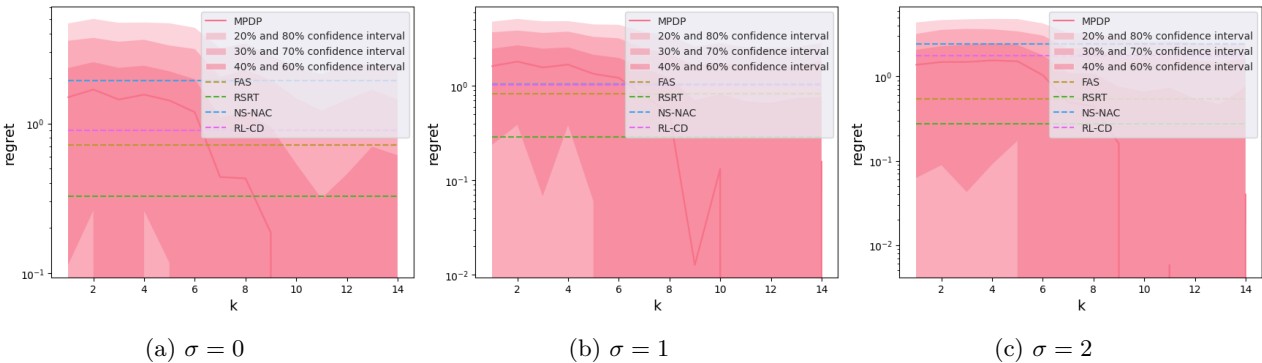

Figure 3: Figure 3a, Figure 3b, and Figure 3c show the regret of MPDP under different additive prediction error $\mathcal{N}(0, \sigma)$ in a setting with a finite number of changes in MDP's. The red solid line shows the mean of the regret, and the shaded area shows the confidence interval.

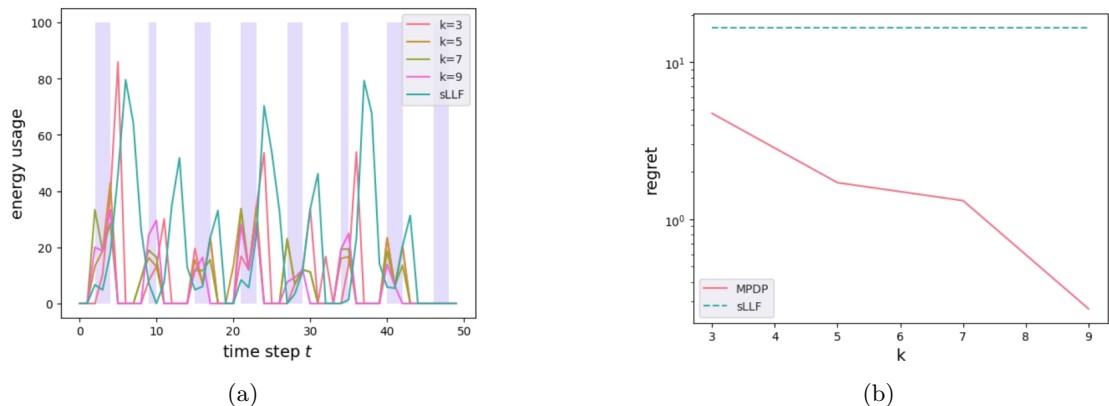

Figure 4: Figure 4a shows the power usage at different time steps. The shaded area indicates the time period with energy prices below 8. We see that with $k \geq 7$, the most energy usage happens within the area with low energy prices, reducing the total energy cost of the station. Figure 4b demonstrates that the regret of EV charging decays with the prediction horizon. Compared with traditional scheduling policies, the proposed algorithm can lower the total energy cost even with a few prediction steps.

exploration probability for every action. Due to the scarcity of changes, both RL-CD and NS-NAC perform worse than RSRT, which is provably optimal for queueing systems with constant $\lambda$. However, the proposed algorithm still outperforms RSRT with $k \geq 9$ in all three noise settings with $\sigma \in \{0, 1, 2\}$, demonstrating the effectiveness of MPDP when model prediction is available.

## 6.2 EV charging

In this section, we consider a scenario of EV charging station under the setup of Example 2 with time horizon $T = 50$. The charging station has three charging stands, and the energy price fluctuates between 2 and 18.

We first show the correlation between the change of energy price and energy usage. We compare the regret of our algorithm with the benchmark policy, the smoothed least-laxity-first algorithm (sLLF) proposed in Chen et al. (2022), which prioritizes charging the EV that is the closest to the departure time. However, given the fluctuating energy price, the optimal policy should charge the EVs at the time steps with the lowest energy price that can still satisfy the energy demand of the EVs before their departure times. As shown in Figure 4a, compared with sLLF, our algorithm selects better time for charging each EV. In particular, when $k$ increases, most of the peak of energy demand falls within the shaded area with low energy price.

We then show the decay of regret with respect to the growth in the prediction horizon. Compared with traditional scheduling algorithm proposed in Chen et al. (2022), our algorithm can better handle the fluctuation in energy price. As shown in Figure 4b, even with only a few steps of prediction, the station's regret decays exponentially.

## 7 Conclusion

This paper designs a noval algorithm for non-stationary MDP utilizing exogenous prediction. We showed, under the assumption of uniform ergodicity, our algorithm achieves a regret of $O(T\gamma^{\lfloor k/J \rfloor}D)$. When $k = O(\log T)$, we obtain a regret sublinear in $T$. We also show that even when the prediction error grows subexponentially, the regret does not explode. The future directions of this work includes the application of this framework in partially observable MDPs and the extension of this framework when only part of the transitional probability and reward functions are predictable.

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

## A    Contraction in span semi-norm

In this section, we briefly introduce some properties of *span* semi-norm, and how it helps to show the contraction property in Theorem 4.1.

**Proposition A.1.** *The span has the following properties:*

1. $\mathrm{sp}(v) \geq 0, \forall v \in \mathbb{R}^d$.

2. $\mathrm{sp}(u + v) \leq \mathrm{sp}(u) + \mathrm{sp}(v)$ *for all* $u, v \in \mathbb{R}^d$.

3. $\mathrm{sp}(kv) = |k| \, \mathrm{sp}(v)$ *for all* $k \in \mathbb{R}, v \in\in \mathbb{R}^d$.

4. $\mathrm{sp}(v + ke) = \mathrm{sp}(v)$ *for all* $k \in \mathbb{R}$*, where* $e = [1, \ldots, 1]^\top$.

5. $\mathrm{sp}(v) = \mathrm{sp}(-v)$.

6. $\mathrm{sp}(v) \leq 2 \|v\|_2$.

The proof of the above proposition easily follows from Definition 1. More detailed properties of the span semi-norm can be found in Puterman (1994). In the following proposition, we show the fundamental step of $J$-stage contraction.

**Proposition A.2.** *Let* $v \in \mathbb{R}^n$ *and policies* $\pi_t, \ldots, \pi_{t+J}$*, then*

$$\mathrm{sp}(P_{t:t+J}^{\pi_t:\pi_{t+J}} v) \leq \gamma \, \mathrm{sp}(v),$$

*where* $P_t^\pi$ *is the transitional probability at time* $t$ *with action determined by* $\pi$*,* $P_{t:t'}^{\pi_t:\pi_{t'}} = P_t^{\pi_t} \cdots P_{t'}^{\pi_{t'}}$*, and*

$$\gamma = 1 - \min_{s,u \in \mathcal{S}} \sum_{j \in \mathcal{S}} \min\{P_{t:t+J}^{\pi_t:\pi_{t+J}}(j|s), P_{t:t+J}^{\pi_t:\pi_{t+J}}(j|u)\}$$

$$= \frac{1}{2} \max_{s,u \in \mathcal{S}} \sum_{j \in \mathcal{S}} |P_{t:t+J}^{\pi_t:\pi_{t+J}}(j|s) - P_{t:t+J}^{\pi_t:\pi_{t+J}}(j|u)| = \max_{s,u \in \mathcal{S}} \sum_{j \in \mathcal{S}} [P_{t:t+J}^{\pi_t:\pi_{t+J}}(j|s) - P_{t:t+J}^{\pi_t:\pi_{t+J}}(j|u)]^+.$$

*Furthermore,* $\gamma \in [0, 1]$*, and there exists* $v$ *such that* $\mathrm{sp}(P_{t:t+J}^{\pi_t:\pi_{t+J}} v) = \gamma \, \mathrm{sp}(v)$.

*Proof:*    For simplicity, we drop subscript $t : t+J$, superscript $a_t : a_{t+J}$, and use $P$ to represent an arbitrary transitional probability matrix. We further define

$$\Lambda(v) = \min_{s \in \mathcal{S}} v(s), \qquad \Upsilon(v) = \max_{s \in \mathcal{S}} v(s).$$

Let $b(i, k; j) := \min\{P(j|i), P(j|k)\}$. For any $v$,

$$\sum_{j \in \mathcal{S}} P(j|i)v(j) - \sum_{j \in \mathcal{S}} P(j|k)v(j)$$

$$= \sum_{j \in \mathcal{S}} [P(j|i) - b(i, k; j)]v(j) - \sum_{j \in \mathcal{S}} [P(j|k) - b(i, k; j)]v(j)$$

$$\leq \sum_{j \in \mathcal{S}} [P(j|i) - b(i, k; j)]\Upsilon(v) - \sum_{j \in \mathcal{S}} [P(j|k) - b(i, k; j)]\Lambda(v)$$

$$= (1 - \sum_{j \in \mathcal{S}} b(i, k; j)) \, \mathrm{sp}(v).$$

Therefore,

$$\mathrm{sp}(P_{t:t+J}^{\pi_t:\pi_{t+J}} v) \leq \max_{i,k \in \mathcal{S}} [1 - \sum_{j \in \mathcal{S}} b(i, k; j)] \, \mathrm{sp}(v),$$

from which the proposition statement immediately follows.    □

We are now ready to prove Proposition 3.1.

*Proof:* [Proof of Proposition 3.1] Let $v^*$ denote the optimal value function at time $t+J+1$, and let $v \in \mathbb{R}^{|S|}$ denote the value function for an arbitrary policy at time $t+J+1$. $s^* = \arg\max_{s \in \mathcal{S}}\{L_{t:t+J}v^*(s) - L_{t:t+J}v(s)\}$, and $s_* = \arg\min_{s \in \mathcal{S}}\{L_{t:t+J}v^*(s) - L_{t:t+J}v(s)\}$. Further, let $\pi^*_{t+i}$ denote the optimal policy according to $L_{t+i} \circ \cdots \circ L_{t+J}(v^*)$, and let $\pi_{t+i}$ denote the optimal policy to take according to $L_{t+i} \circ \cdots \circ L_{t+J}(v)$ then

$$L_{t:t+J}v^*(s^*) - L_{t:t+J}v(s^*) \leq L_{t:t+J}^{\pi^*_{t:t+J}}v^*(s^*) - L_{t:t+J}^{\pi^*_{t:t+J}}v(s^*) = P_{t:t+J}^{\pi^*_{t:t+J}}(v^* - v)(s^*),$$

$$L_{t:t+J}v^*(s_*) - L_{t:t+J}v(s_*) \geq L_{t:t+J}^{\pi^2_{t:t+J}}v^*(s_*) - L_{t:t+J}^{\pi_{t:t+J}}v(s_*) = P_{t:t+J}^{\pi_{t:t+J}}(v^* - v)(s_*).$$

Therefore,

$$\begin{aligned}
\mathrm{sp}(L_{t:t+J}v^* - L_{t:t+J}v) &\leq P_{t:t+J}^{\pi^*_{t:t+J}}(v^* - v)(s^*) - P_{t:t+J}^{\pi_{t:t+J}}(v^* - v)(s_*) \\
&\leq \max_{s \in \mathcal{S}} P_{t:t+J}^{\pi^*_{t:t+J}}(v^* - v)(s) - \min_{s \in \mathcal{S}} P_{t:t+J}^{\pi_{t:t+J}}(v^* - v)(s) \\
&\leq \mathrm{sp}\left(\begin{bmatrix} P_{t:t+J}^{\pi^*_{t:t+J}} \\ P_{t:t+J}^{\pi_{t:t+J}} \end{bmatrix}(v^* - v)\right).
\end{aligned} \tag{16}$$

Applying Proposition A.2 to equation 16 immediately leads to the theorem statement. $\qquad\square$

## B  Diameter

In this section, we prove Proposition 3.3, which shows that the span semi-norm of value function $V_t$ is upper bounded by diameter $D$ for all $t$.

*Proof:* [Proof of Proposition 3.3] Let $\boldsymbol{\pi}^*$ denote the optimal policy and $\boldsymbol{\pi}'$ denote the policy defined in Definition 3 trying to move fastest to $\{s^*_{t+i}\}_{i \in [D]}$. By Definition 3, under a trajectory generated by $\boldsymbol{\pi}'$ starting from time $t$ at $s_*$, define $d = \inf\{\tau : \tau > 0, s_{t+\tau} = s^*_{t+\tau}\}$. $d$ is a stopping time and by Definition 3, $\mathbb{E}[d] \leq D$.

$$\begin{aligned}
\mathrm{sp}(V_t) =& V_t^*(s^*) - V_t^*(s_*) \\
=& \mathbb{E}^{\boldsymbol{\pi}^*}\left[\sum_{h=t}^{t+d-1} r_h(s_h, a_h) + V_{t+d}^*(s_{t+d})|s_t = s^*\right] - \mathbb{E}^{\boldsymbol{\pi}^*}\left[\sum_{h=t}^{t+d-1} r_h(s_h, a_h) + V_{t+d}^*(s_{t+d})|s_t = s_*\right] \\
\leq& \mathbb{E}^{\boldsymbol{\pi}^*}\left[\sum_{h=t}^{t+d-1} r_h(s_h, a_h) + V_{t+d}^*(s_{t+d})|s_t = s^*\right] - \mathbb{E}^{\boldsymbol{\pi}'}\left[\sum_{h=t}^{t+d-1} r_h(s_h, a_h) + V_{t+d}^*(s_{t+d})|s_t = s_*\right] \\
=& \left(\mathbb{E}^{\boldsymbol{\pi}^*}\left[\sum_{h=t}^{t+d-1} r_h(s_h, a_h)|s_t = s^*\right] - \mathbb{E}^{\boldsymbol{\pi}'}\left[\sum_{h=t}^{t+d-1} r_h(s_h, a_h)|s_t = s_*\right]\right) \\
&+ \underbrace{\left(\mathbb{E}^{\boldsymbol{\pi}^*}[V_{t+d}^*(s_{t+d})|s_t = s^*] - V_{t+d}^*(s^*_{t+d})\right)}_{<0} \\
\leq& \mathbb{E}d \leq D.
\end{aligned}$$

$\qquad\square$

Intuitively, if $V_t^*(s^*) > V_t^*(s_*)$, then a better policy will be to move to $\arg\max_s V_{t+D}^*(s)$ as fast as possible, during which only $D$ reward will be lost in expectation.

## C Proof of Lemma 5.2

In this section, we prove Lemma 5.2, which bounds the step-wise error incurred by using the forecast system dynamics, instead of the ground-truth transition and reward functions.

*Proof:* [Proof of Lemma 5.2] Assume that $\text{sp}\left(\tilde{V} - \hat{V}\right) < b$. Given a state $s$, let $\tilde{a}$ denote the action chosen in $L_{t+\ell}\tilde{V}(s)$, and $\hat{a}$ denote the action chosen in $\hat{L}_{t+\ell|t}\hat{V}(s)$. By the construction of Algorithm 1, we obtain

$$\hat{r}_{t+\ell|t}(s, \hat{a}) + \mathbb{E}_{s' \sim \hat{P}_{t+\ell|t}(\cdot|s,\hat{a})} \hat{V}(s') \geq \hat{r}_{t+\ell|t}(s, \tilde{a}) + \mathbb{E}_{s' \sim \hat{P}_{t+\ell|t}(\cdot|s,\tilde{a})} \hat{V}(s')$$

$$\geq r_{t+\ell}(s, \tilde{a}) - \epsilon_\ell + \mathbb{E}_{s' \sim P_{t+\ell}(\cdot|s,\tilde{a})} \hat{V}(s') - \left( \mathbb{E}_{s' \sim P_{t+\ell}(\cdot|s,\tilde{a})} \hat{V}(s') - \mathbb{E}_{s' \sim \hat{P}_{t+\ell|t}(\cdot|s,\tilde{a})} \hat{V}(s') \right) \qquad (17)$$

$$\geq r_{t+\ell}(s, \tilde{a}) - \epsilon_\ell + \mathbb{E}_{s' \sim P_{t+\ell}(\cdot|s,\tilde{a})} \tilde{V}(s') - \mathbb{E}_{s' \sim P_{t+\ell}(\cdot|s,\tilde{a})} \left( \tilde{V}(s') - \hat{V}(s') \right) - \delta_\ell D.$$

The first inequality is due to the relative optimality of $\hat{a}$ for $\hat{V}$, and the second/third inequalities are by Assumption 1 and Holder's inequality. Similarly,

$$r_{t+\ell}(s, \tilde{a}) + \mathbb{E}_{s' \sim P_{t+\ell}(\cdot|s,\tilde{a})} \tilde{V}(s') \geq r_{t+\ell}(s, \hat{a}) + \mathbb{E}_{s' \sim P_{t+\ell}(\cdot|s,\hat{a})} \tilde{V}(s')$$

$$\geq \hat{r}_{t+\ell}(s, \hat{a}) - \epsilon_\ell + \mathbb{E}_{s' \sim P_{t+\ell}(\cdot|s,\hat{a})} \hat{V}(s') - \mathbb{E}_{s' \sim P_{t+\ell}(\cdot|s,\hat{a})} \left( \hat{V}(s') - \tilde{V}(s') \right) \qquad (18)$$

$$\geq \hat{r}_{t+\ell}(s, \hat{a}) - \epsilon_\ell + \mathbb{E}_{s' \sim \hat{P}_{t+\ell|t}(\cdot|s,\hat{a})} \hat{V}(s') - \mathbb{E}_{s' \sim P_{t+\ell}(\cdot|s,\hat{a})} \left( \hat{V}(s') - \tilde{V}(s') \right) - \delta_\ell D.$$

Combining equation 17 and equation 18, we obtain

$$r_{t+\ell}(s, \tilde{a}) + \mathbb{E}_{s' \sim P_{t+\ell}(\cdot|s,\tilde{a})} \tilde{V}(s') - \mathbb{E}_{s' \sim P_{t+\ell}(\cdot|s,\tilde{a})} \left( \tilde{V}(s') - \hat{V}(s') \right) - \epsilon_\ell - \delta_\ell D$$

$$\leq \hat{r}_{t+\ell|t}(s, \hat{a}) + \mathbb{E}_{s' \sim \hat{P}_{t+\ell|t}(\cdot|s,\hat{a})} \hat{V}(s') \qquad (19)$$

$$\leq r_{t+\ell}(s, \tilde{a}) + \mathbb{E}_{s' \sim P_{t+\ell}(\cdot|s,\tilde{a})} \tilde{V}(s') - \mathbb{E}_{s' \sim P_{t+\ell}(\cdot|s,\hat{a})} \left( \tilde{V}(s') - \hat{V}(s') \right) + \epsilon_\ell + \delta_\ell D.$$

Therefore,

$$\hat{L}_{t+\ell|t}\hat{V}(s) - L_{t+\ell}\tilde{V}(s) = \hat{r}_{t+\ell|t}(s, \hat{a}) + \mathbb{E}_{s' \sim \hat{P}_{t+\ell|t}(\cdot|s,\hat{a})} \hat{V}(s') - r_{t+\ell}(s, \tilde{a}) - \mathbb{E}_{s' \sim P_{t+\ell}(\cdot|s,\tilde{a})} \tilde{V}(s'). \qquad (20)$$

Substituting equation 20 into equation 19, we obtain

$$-\mathbb{E}_{s' \sim P_{t+\ell}(\cdot|s,\tilde{a})} \left( \tilde{V}(s') - \hat{V}(s') \right) - \epsilon_\ell - \delta_\ell D \leq \hat{L}_{t+\ell|t}\hat{V}(s) - L_{t+\ell}\tilde{V}(s) \leq -\mathbb{E}_{s' \sim P_{t+\ell}(\cdot|s,\hat{a})} \left( \tilde{V}(s') - \hat{V}(s') \right) + \epsilon_\ell + \delta_\ell D. \qquad (21)$$

Let $s^* := \arg\max_s \{\hat{L}_{t+\ell|t}\hat{V}(s) - L_{t+\ell}\tilde{V}(s)\}$ and $s_* := \arg\min_s \{\hat{L}_{t+\ell|t}\hat{V}(s) - L_{t+\ell}\tilde{V}(s)\}$. Using Definition 1 and the assumption that $\text{sp}\left(\tilde{V} - \hat{V}\right) < b$, we obtain

$$\text{sp}(\hat{L}_{t+\ell|t}\hat{V} - L_{t+\ell}\tilde{V}) = \hat{L}_{t+\ell|t}\hat{V}(s^*) - L_{t+\ell}\tilde{V}(s^*) - \left( \hat{L}_{t+\ell|t}\hat{V}(s_*) - L_{t+\ell}\tilde{V}(s_*) \right)$$

$$\leq \|P_{t+\ell}(\cdot|s^*, \hat{a}) - P_{t+\ell}(\cdot|s_*, \tilde{a})\|_{\text{TV}} \, \text{sp}(\tilde{V} - \hat{V}) + \epsilon_\ell + \delta_\ell D + \epsilon_\ell + \delta_\ell D \qquad (22)$$

$$\leq b + 2\epsilon_\ell + 2\delta_\ell D.$$

$\square$

## D Bounding the error of $Q$ function

In order to bound the error of $Q$ function for every $t$, we must first bound the error of each estimated value function $\hat{\psi}_t^0$. We complete this in two steps. First, we bound the difference between $\hat{\psi}_t^0$ and $\tilde{\psi}_t^0$ in span semi-norm. Then, we bound the difference between $\tilde{\psi}_t^0$ and $V_t^*$.

**Proposition D.1.**

$$\mathrm{sp}\left(\hat{\psi}_t^0 - \tilde{\psi}_t^0\right) \leq 2 \sum_{i=0}^{\lceil k/J \rceil - 1} \gamma^i \left(\sum_{j=1}^{J} \epsilon_{iJ+j} + \sum_{j=1}^{J} \delta_{iJ+j} D\right) + 2\gamma^{\lfloor k/J \rfloor} \left(\sum_{j=1}^{k\%J} \epsilon_{\lfloor k/J \rfloor \cdot J + j} + \sum_{j=1}^{k\%J} \delta_{\lfloor k/J \rfloor \cdot J + j} D\right),$$

*for all t, where $k\%J := k - \lfloor k/J \rfloor \cdot J$.*

*Proof:* We prove the case where $t + k \leq T$, the case where $t + k > T$ follows the exact same line. By Lemma 5.1,

$$\tilde{\psi}_t^0 - \hat{\psi}_t^0 = L_t \circ \cdots \circ L_{t+k} W_0 - \hat{L}_{t|t} \circ \cdots \circ \hat{L}_{t+k|t} W_0. \tag{23}$$

We proceed step-wise on $\mathrm{sp}(\hat{\psi}_t^\ell - \tilde{\psi}_t^\ell)$ for $\ell$ going backward from $k$ to $0$. For the base case where $\ell = k + 1$, we have $\mathrm{sp}\left(\tilde{\psi}_t^{k+1} - \hat{\psi}_t^{k+1}\right) = \mathrm{sp}(W_0 - W_0) = 0$.

By Proposition 3.3, we know $\mathrm{sp}(V_t) \leq D$. By the monotonicity of Bellman operator, $\mathrm{sp}(\hat{\psi}_t^\ell) < D, \mathrm{sp}(\tilde{\psi}_t^\ell) < D$ for all $t, \ell$. Therefore, we obtain

$$\mathrm{sp}\left(\tilde{\psi}_t^0 - \hat{\psi}_t^0\right) = \mathrm{sp}\left(L_t \circ \cdots \circ L_{t+k} W_0 - \hat{L}_t \circ \cdots \circ \hat{L}_{t+k} W_0\right)$$

$$\leq \mathrm{sp}\left(L_t \circ \cdots \circ L_{t+J}\left(\tilde{\psi}_t^J - \hat{\psi}_t^J\right)\right) + 2\sum_{i=1}^{J} \epsilon_i + 2\sum_{i=1}^{J} \delta_i D \tag{24}$$

$$\leq \gamma \, \mathrm{sp}\left(\tilde{\psi}_t^J - \hat{\psi}_t^J\right) + 2\sum_{i=1}^{J} \epsilon_i + 2\sum_{i=1}^{J} \delta_i D \tag{25}$$

$$\vdots$$

$$\leq 2 \sum_{i=0}^{\lceil k/J \rceil - 1} \gamma^i \left(\sum_{j=1}^{J} \epsilon_{iJ+j} + \sum_{j=1}^{J} \delta_{iJ+j} D\right) + 2\gamma^{\lfloor k/J \rfloor} \left(\sum_{j=1}^{k\%J} \epsilon_{\lfloor k/J \rfloor \cdot J + j} + \sum_{j=1}^{k\%J} \delta_{\lfloor k/J \rfloor \cdot J + j} D\right), \tag{26}$$

where equation 24 is obtained by Lemma 5.2, and equation 25 is obtained by Proposition 3.1. We repeat the steps in equation 24 and equation 25 to obtain equation 26. □

We are now ready to bound the difference between $\tilde{\psi}_t^0$ and $V_t^*$ in span semi-norm.

**Lemma D.2.** *Given $t + k < T$,*
$$\mathrm{sp}(\tilde{\psi}_t^0 - V_t^*) \leq \gamma^{k/J} \mathrm{sp}(V_t^*).$$
*If $t + k \geq T$, the estimation of $V_t^*$ is exact, i.e. $\tilde{\psi}_t^k = V_t^*$.*

*Proof:* The latter equality is clear from the definition of the Bellman operator, so we just need to prove the first inequality.

$$\mathrm{sp}(\tilde{\psi}_t^0 - V_t^*) = \mathrm{sp}\left(L_t \circ \cdots \circ L_{t+k} \circ (W_0) - L_t \circ \cdots \circ L_{t+k} \circ \left(V_{t+k+1}^*\right)\right)$$
$$\leq \gamma^{\lfloor k/J \rfloor} \mathrm{sp}\left(W_0 - V_{t+k+1}^*\right)$$
$$= \gamma^{\lfloor k/J \rfloor} \mathrm{sp}\left(V_{t+k+1}^*\right),$$

where we applied Proposition 3.1 for $\lfloor k/J \rfloor$ times. The last equality holds because $W_0 = 0$. □

We are now ready to bound the difference between the optimal $Q$ function $Q_t^*$ and the estimated $Q$ function $\hat{\Psi}_t$. Similar to the beginning of this section, we divide the proof into two steps. First, we bound the difference between the optimal $Q$ function $Q_t^*$ and the estimated $Q$ function with exact forecast $\tilde{\Psi}_t$.

**Lemma D.3.** *For any states $s$ and $t + k < T$, the $Q$ function of the MDP satisfies*

$$\mathrm{sp}(Q_t^*(s,\cdot) - \tilde{\Psi}_t(s,\cdot)) \leq \gamma^{\lfloor k/J \rfloor} \, \mathrm{sp}(V_{t+k+1}^*).$$

*If $t + k \geq T$, the estimation of $Q_t^*$ is exact, i.e. $\tilde{\Psi}_t(s,\cdot) = Q_t^*(s,\cdot)$.*

*Proof:*   The latter equality is a direct result of the construction of $\tilde{\Psi}_t$ in equation 14, so we only need to prove the first inequality. By the Bellman operator,

$$
\begin{aligned}
\mathrm{sp}(Q_t^*(s,\cdot) - \tilde{\Psi}_t(s,\cdot)) = & \max_a \left( (r_t(s,a) + \mathbb{E}[V_{t+1}^*(s_{t+1})|s_t = s, a_t = a]) \right. \\
& \left. - (r_t(s,a) + \mathbb{E}[\tilde{\psi}_{t+1}^0(s_{t+1})|s_t = s, a_t = a]) \right) \\
& - \min_a \left( (r_t(s,a) + \mathbb{E}[V_{t+1}^*(s_{t+1})|s_t = s, a_t = a]) \right. \\
& \left. - (r_t(s,a) + \mathbb{E}[\tilde{\psi}_{t+1}^0(s_{t+1})|s_t = s, a_t = a]) \right) \\
= & \max_a \mathbb{E}[V_{t+1}^*(s_{t+1}) - \tilde{\psi}_{t+1}^0(s_{t+1})|s_t = s, a_t = a]) \\
& - \min_a \mathbb{E}[V_{t+1}^*(s_{t+1}) - \tilde{\psi}_{t+1}^0(s_{t+1})|s_t = s, a_t = a]) \\
\leq & \max_s \mathbb{E}[V_{t+1}^*(s) - \tilde{\psi}_{t+1}^0(s)]) - \min_s \mathbb{E}[V_{t+1}^*(s) - \tilde{\psi}_{t+1}^0(s)]) \\
= & \, \mathrm{sp}(V_{t+1}^* - \tilde{\psi}_{t+1}^0) \\
\leq & \, \gamma^{\lfloor k/J \rfloor} \, \mathrm{sp}\left( V_{t+k+1}^* \right),
\end{aligned}
$$

where we used Lemma D.2 for the last inequality. $\qquad \square$

Then, we bound the difference between the $\tilde{\Psi}_t$ and the estimated $Q$ function $\hat{\Psi}_t$ with forecast under Assumption 1.

**Lemma D.4.** *For any state $s$ and time $t$,*

$$\mathrm{sp}\left( \tilde{\Psi}_t(s,\cdot) - \hat{\Psi}_t(s,\cdot) \right) < 2\epsilon_0 + 2\delta_0 D + 2 \sum_{i=0}^{\lceil k/J \rceil - 1} \gamma^i \left( \sum_{j=1}^{J} \epsilon_{iJ+j} + \sum_{j=1}^{J} \delta_{iJ+j} D \right) + 2\gamma^{\lfloor k/J \rfloor} \left( \sum_{j=1}^{k\%J} \epsilon_{\lfloor k/J \rfloor \cdot J + j} + \sum_{j=1}^{k\%J} \delta_{\lfloor k/J \rfloor \cdot J + j} D \right).$$

*Proof:*   By triangle inequality in Proposition A.1, we have

$$
\begin{aligned}
\mathrm{sp}\left( \tilde{\Psi}_t(s,\cdot) - \hat{\Psi}_t(s,\cdot) \right) \leq & \, \mathrm{sp}\left( r_t(s,\cdot) - \hat{r}_{t|t}(s,\cdot) \right) + \mathrm{sp}\left( \mathbb{E}_{s' \sim P_t(\cdot|s,\cdot)}[\tilde{\psi}_{t+1}^0(s')] - \mathbb{E}_{s' \sim \hat{P}_t(\cdot|s,\cdot)}[\hat{\psi}_{t+1}^0(s')] \right) \\
\leq & \, 2\epsilon_0 + \mathrm{sp}\left( \mathbb{E}_{s' \sim P_t(\cdot|s,\cdot)}[\tilde{\psi}_{t+1}^0(s') - \hat{\psi}_{t+1}^0(s')] \right) + \mathrm{sp}\left( \mathbb{E}_{s' \sim P_t(\cdot|s,\cdot)}[\hat{\psi}_{t+1}^0(s')] - \mathbb{E}_{s' \sim \hat{P}_t(\cdot|s,\cdot)}[\hat{\psi}_{t+1}^0(s')] \right) \\
\leq & \, 2\epsilon_0 + \mathrm{sp}\left( \mathbb{E}_{s' \sim P_t(\cdot|s,\cdot)}[\tilde{\psi}_{t+1}^0(s') - \hat{\psi}_{t+1}^0(s')] \right) + 2\delta_0 D \\
\leq & \, 2\epsilon_0 + 2\delta_0 D + 2 \sum_{i=0}^{\lceil k/J \rceil - 1} \gamma^i \left( \sum_{j=1}^{J} \epsilon_{iJ+j} + \sum_{j=1}^{J} \delta_{iJ+j} D \right) + 2\gamma^{\lfloor k/J \rfloor} \left( \sum_{j=1}^{k\%J} \epsilon_{\lfloor k/J \rfloor \cdot J + j} + \sum_{j=1}^{k\%J} \delta_{\lfloor k/J \rfloor \cdot J + j} D \right),
\end{aligned}
$$

where we used Proposition D.1 in the last inequality. $\qquad \square$

We are now ready for Proof of Corollary 5.3.

*Proof:*   [Proof of Corollary 5.3] Fix a state $s$ and time $t$. Let $a^*$ denote an optimal action at $(t, s)$, and let $a$ be the action chosen by Algorithm 1. Define the error vector over actions

$$e_t(a') := \hat{\Psi}_t(s, a') - Q_t^*(s, a'), \qquad a' \in \mathcal{A}.$$

We first relate the suboptimality gap to the span of this error:

$$Q_t^*(s, a^*) - Q_t^*(s, a) = \hat{\Psi}_t(s, a^*) - e_t(a^*) - \left( \hat{\Psi}_t(s, a) - e_t(a) \right)$$

$$= \big(\hat{\Psi}_t(s, a^*) - \hat{\Psi}_t(s, a)\big) + \big(e_t(a) - e_t(a^*)\big).$$

Since $a$ maximizes $\hat{\Psi}_t(s, \cdot)$, we have $\hat{\Psi}_t(s, a) \geq \hat{\Psi}_t(s, a^*)$ and hence $\hat{\Psi}_t(s, a^*) - \hat{\Psi}_t(s, a) \leq 0$. Therefore

$$Q_t^*(s, a^*) - Q_t^*(s, a) \leq e_t(a) - e_t(a^*)$$
$$\leq \max_{a', a''} \big(e_t(a') - e_t(a'')\big) = \mathrm{sp}(e_t).$$

That is,

$$Q_t^*(s, a^*) - Q_t^*(s, a) \leq \mathrm{sp}\big(\hat{\Psi}_t(s, \cdot) - Q_t^*(s, \cdot)\big). \tag{27}$$

Next, use the triangle inequality for the span seminorm (Proposition A.1):

$$\mathrm{sp}\big(\hat{\Psi}_t(s, \cdot) - Q_t^*(s, \cdot)\big) \leq \mathrm{sp}\big(\hat{\Psi}_t(s, \cdot) - \tilde{\Psi}_t(s, \cdot)\big) + \mathrm{sp}\big(\tilde{\Psi}_t(s, \cdot) - Q_t^*(s, \cdot)\big). \tag{28}$$

The second term in equation 28 is controlled by Lemma D.3:

$$\mathrm{sp}\big(\tilde{\Psi}_t(s, \cdot) - Q_t^*(s, \cdot)\big) = \mathrm{sp}\big(Q_t^*(s, \cdot) - \tilde{\Psi}_t(s, \cdot)\big) \leq \gamma^{\lfloor k/J \rfloor} \mathrm{sp}\big(V_{t+k+1}^*\big). \tag{29}$$

The first term in equation 28 is bounded by Lemma D.4:

$$\mathrm{sp}\big(\hat{\Psi}_t(s, \cdot) - \tilde{\Psi}_t(s, \cdot)\big) \leq 2\epsilon_0 + 2\delta_0 D$$
$$+ 2 \sum_{i=0}^{\lceil k/J \rceil - 1} \gamma^i \bigg( \sum_{j=1}^{J} \epsilon_{iJ+j} + \sum_{j=1}^{J} \delta_{iJ+j} D \bigg)$$
$$+ 2\gamma^{\lfloor k/J \rfloor} \bigg( \sum_{j=1}^{k\%J} \epsilon_{\lfloor k/J \rfloor J+j} + \sum_{j=1}^{k\%J} \delta_{\lfloor k/J \rfloor J+j} D \bigg). \tag{30}$$

Combining equation 27, equation 28, equation 29 and equation 30, we obtain

$$Q_t^*(s, a^*) - Q_t^*(s, a) \leq \gamma^{\lfloor k/J \rfloor} \mathrm{sp}\big(V_{t+k+1}^*\big) + 2\epsilon_0 + 2\delta_0 D$$
$$+ 2 \sum_{i=0}^{\lceil k/J \rceil - 1} \gamma^i \bigg( \sum_{j=1}^{J} \epsilon_{iJ+j} + \sum_{j=1}^{J} \delta_{iJ+j} D \bigg)$$
$$+ 2\gamma^{\lfloor k/J \rfloor} \bigg( \sum_{j=1}^{k\%J} \epsilon_{\lfloor k/J \rfloor J+j} + \sum_{j=1}^{k\%J} \delta_{\lfloor k/J \rfloor J+j} D \bigg).$$

$$\square$$

**Theorem D.5.** *If* $Q_t^*(s, a_t^*) - \max_{a, a \neq a_t^*} \{Q_0^*(s, a)\} > \gamma^{\lfloor k/J \rfloor} \mathrm{sp}\big(V_{t+k}^*\big)$ *for all* $t, s$, *then the proposed algorithm is equivalent to the optimal policy.*

# E   Proof of Theorem 4.1

We are now ready to prove Theorem 4.1.

*Proof:* [Proof of Theorem 4.1] Let $\{(\tilde{s}_i, \tilde{a}_i)\}_i$ denote the sequence of state-action pairs generated by Algorithm 1 with accurate prediction , and $\{s_i, a_i\}$ denote the sequence of state-action pairs generated by Algorithm 1. Let $\{a_i^*\} := \{\arg\max_a Q_i(s_i, a)\}_i$ denote the optimal action at each of those states.

$$V_0^*(s_0) - V_0(s_0) = (r_0(s_0, a_0^*) - r_0(s_0, a_0)) + (\mathbb{E}_{s_1 \sim P(\cdot|s_0, a_0^*)}[V_1^*(s_1)] - \mathbb{E}_{s_1 \sim P(\cdot|s_0, a_0)}[V_1^*(s_1)])$$
$$+ (\mathbb{E}_{s_1 \sim P(\cdot|s_0, a_0)}[V_1^*(s_1) - V_1(s_1)]) \tag{31}$$
$$= (Q_0^*(s_0, a_0^*) - Q_0^*(s_0, a_0)) + \mathbb{E}_{s_1 \sim P(\cdot|s_0, \tilde{a}_0)}[V_1^*(s_1) - V_1(s_1)] \tag{32}$$

$$\leq \sum_{t=0}^{T-k-2} \gamma^{\lfloor k/J \rfloor} \operatorname{sp}\left(V_{t+k+2}^*\right) + 2T\epsilon_0 + 2T\delta_0 D + 2T \sum_{i=0}^{\lceil k/J \rceil - 1} \gamma^i \left( \sum_{j=1}^{J} \epsilon_{iJ+j} + \sum_{j=1}^{J} \delta_{iJ+j} D \right) \tag{33}$$

$$+ 2T\gamma^{\lfloor k/J \rfloor} \left( \sum_{j=1}^{k\%J} \epsilon_{\lfloor k/J \rfloor \cdot J + j} + \sum_{j=1}^{k\%J} \delta_{\lfloor k/J \rfloor \cdot J + j} D \right). \tag{34}$$

In equation 31 and equation 32, we expand out the value functions and rearrange the terms. Applying Corollary 5.3 leads to the resulting bound. □

## F  Lower bound of regret for non-stationary MDP not satisfying Assumption 2

In order to show the necessity of Assumption 2, we provide the following counter-example that generates a linear regret for any $k$.

*Counter-example* F.1. For any fixed $k, T$ such that $k \ll T$, we can generate the following non-stationary MDP with three states $\{s^i\}_{i=1,2,3}$. The learner starts at $s^1$ and must make the decision of going to $s^2$ or $s^3$ at time step 0. Each of these actions has a reward of 0. $s^2, s^3$ are sinks, and the learner can not move out after time step 1. Clearly, the above MDP does not satisfy Assumption 2, since for any $J > 0$, $P_t^{\pi_t} \circ \cdots P_{t+J}^{\pi_{t+J}}(s^1|s^2) = 0$ for all policies $\boldsymbol{\pi}$. At time step 0, a state $s^* \in \{s_2, s_3\}$ is chosen at random, such that the learner gets a reward of 1 if at $s^*$ for all time steps after $k+1$, and a reward of 0 if otherwise. We claim that any algorithm with prediction horizon $k$ would generate a linear regret for the above MDP, as there is a non-zero constant probability for any policy to not choose $s^*$ at time step 0.

