# OpenReview forum: "Predictive Control and Regret Analysis of Non-Stationary MDP with Look-ahead Information"
_TMLR — Accepted by TMLR_

### Review · Reviewer_vuXT · 2025-03-10

**Summary Of Contributions:**

The authors consider a very interesting topic -- how we can design policy in non-stationary MDP with some forecasts. The results are quite strong as the regret decreases exponentially. Also, they used both simulation and real-world data to show the advantages of their proposed algorithm.

**Audience:**

Yes

**Broader Impact Concerns:**

I don't think it has broader impact concerns.

**Claims And Evidence:**

Yes

**Requested Changes:**

1. Add more comparison on your Assumption 1 and other budget assumptions.
2. Add some discussion on how $D$ depends on $T$ and the reason why you don't need exploration.
3. Try to find a more delicate assumption on $\epsilon_l$ and $\delta_l$.

**Strengths And Weaknesses:**

The paper is interesting and technically strong. However, I still have some questions on the assumptions and intuitions of the algorithm. First, for Definition 1, is there any difference between your assumption and the budget assumption in the previous literature? Intuitively, since the size of data is increasing, we can get more and more accurate estimations. And it is usually equivalent to the case that we have a limited non-stationary budget. For Definition 3, it seems more reasonable for me that $\alpha$ then $\gamma$ also depends on $J$. Can you give any example that $\gamma$ is independent of $J$? For Definition 4, in the worst case, $D$ can be very large, even linear to $T$. Can you add more discussion on $D$? For (10), you only use a greedy algorithm without exploration. This kind of algorithm usually doesn't work unless you have very strong assumptions on parameters, like $D$. At the same time, the computational complexity is exponential to $k$, making it hard to use in practice. For Theorem 4.1, since the assumption on $\epsilon_0$ should be held for all $t$, considering the case that we don't have enough data at the beginning, $\epsilon_0$ can be $\Omega(1)$. It will make the regret bound trivial. In the discussion of Corollary 4.2, in some RL literature, even without predictions, it's feasible to get sublinear regret. What is the unique difficulty of your problem?

---

> ### Author Response · Authors · 2025-03-17
>
> We would like to thank the reviewer for taking the time to read the paper and provide some clarifications and discussions below.
>
> - Definition 1 diverges from previous studies [Merlis(2024), Lee et. al. (2024)] in that Definition 1 gives a bound on future system dynamics prediction, so we do not need to place an assumption on variation budget. The variation budget is typically defined as $\sum_{t=1}^{T-1} \Vert P_t - P_{t-1}\Vert + \Vert r_t - r_{t-1}\Vert < \Delta_T$, which would be large if the rewards or transitional probabilities fluctuates during the learning process [Wei \& Luo (2021); Jali et. al., (2025)].  The reviewer's comment that the data size would increase the accuracy of system estimation is indeed very true. However, this estimation would always be based on the past/current system dynamics. If the variation budget is very high, the system dynamics in the near future can diverge significantly from the current system dynamics, generating high regret. However, even in those cases, we can still have a relatively accurate estimation of the near future, enabling the proposed algorithm to generate a far lower regret.
> - The reviewer is correct that $\alpha$ and $\gamma$ does depend on $J$. In this particular case, we treat each as a constant for any particular MDP. Clearly, in most MDPs, there might not exist $\gamma < 1$ for a very small $J$. However, as $J$ increases, any ergotic MDP would have a $\gamma < 1$, which will decrease exponentially as $J$ increases. In this particular work, we just pick the minimal $J$ such that $\gamma < 1$, and treat each of them as constant.
> - The definition of the diameter does imply that, in the worst case, it is as large as $T$. However, this implies that there exist certain states and certain time steps such that if the agent takes the wrong action, it can not return to the optimal trajectory of states before the end of the time horizon in expectation, regardless of the policy used, which inevitably generates a linear regret for any policy with the same prediction horizon and accuracy. In this case, a large diameter would be the result of the disconnectivity of the MDP, which affects all policies uniformly. In practice, such as in the server and EV-charging examples, the optimal states are concentrated in a small subset of the overall set that minimizes the cost, and the overall state dynamics are ergodic. In those cases, the diameter would be small.
> - Equation (10) indeed looks exponential in $k$ if implemented as is. However, as we use a dynamic-programming-style algorithm, the complexity is polynomial in $k$. We can use the greedy dynamic programming algorithm for action generation yet still achieve a sublinear regret bound because the system forecast with accuracy bound itself is, in a way, an exploration into all possible developments of the system in the near future. By optimizing the predicted future dynamics, we are also minimizing the regret experienced by the agent. A better exploration would involve a longer prediction horizon $k$ and lower prediction error in this setting. In extreme cases, if the agent has a prediction horizon $k=T$ and no prediction error, it would be able to develop the optimal policy by definition, implying the system is fully explored.
> - If $\epsilon_0$ is large, this algorithm will face issues with regret. However, in many practical cases, such as weather forecast and energy forecast, we have $\epsilon_0 = 0$, as the weather and energy production/usage at the current moment is known exactly to the agent. In those cases, the proposed algorithm would produce very low regret.
>
> In addition to the above, we updated the related work section to better reflect the distinction between this work and the theoretical bound provided by time-varying RL. We also added a time-varying RL algorithm [Jali et. al. (2025)] as a benchmark in the simulation section.  Lastly, we rephrased some of the examples and definitions for clarification.
> We hope that we have clarified any confusion above. Please let us know if you have any further questions.

---

### Review · Reviewer_KuNg · 2025-03-18

**Summary Of Contributions:**

This paper studies policy learning in non-stationary Markov Decision Processes (MDPs), which are usually hard to solve due to their time-varying transition and reward functions. It proposes to utilize look-ahead predictions in dynamic programming to determine the optimal action. The authors prove theoretically that the regret decreases exponentially as the length of the look-ahead window increases. Moreover, as long as the prediction error grows sub-exponentially with the prediction horizon, the regret does not explode.

**Audience:**

Yes

**Claims And Evidence:**

Yes

**Requested Changes:**

- External information like look-ahead predictions can help policy learning in unknown non-stationary MDPs. However, a key problem with this approach is how to obtain the predictions, especially sufficiently accurate ones (e.g., predictions with sub-exponentially growing prediction error). In the first simulation setting regarding queueing systems, the prediction is directly set to the true value plus some Gaussian noise. In the second simulation setting concerning EV charging, the prediction setup is not explicitly discussed (if I did not miss anything). Example 2 only mentions that "both weather and energy demand can be forecasted." In either case, it is unclear how accurately the predictions can be constructed in real-world problems. It would be helpful for the authors to provide a real-world example of the predictions.
- Could the authors explain why the confidence intervals in Figure 1 are wide? The proposed method does not seem to be significantly better than the baselines, even when $\sigma = 0$.

Some minor questions:
- This paper proves that theoretically, when the look-ahead window $k = O(log T)$, the regret is sublinear in $T$. In the related work section, it would be helpful if the authors could discuss the regret bounds in the literature for non-stationary MDPs and under which assumptions they are derived.
- Why do the arrival rates of jobs fluctuate periodically in Figure 1a? Is it part of the problem setup or a result of the algorithm?
- What is the total time horizon $T$ in the two examples in Section 6?
- Does the contraction coefficient $\gamma$ refer to $\alpha$ in Definition 3?
- In Definition 4, should $s'$ in $\mathcal{T} (s' | \pi, s)$ be $\{S^{(i)}\}_i$?

**Strengths And Weaknesses:**

A non-stationary MDP is intrinsically hard to solve since we have limited data to learn about the non-stationary process at each time point. Exogenous predictions provide external information that can help in learning optimal policies. The paper rigorously bounds the regret in terms of the diameter of a non-stationary MDP and the $J$-stage contraction coefficient. It is also interesting to see a theoretical demonstration that the regret does not explode under mild conditions, and that numerically the regret grows slowly with the growth rate $g$.

However, the assumptions under which the proposed method performs well should be further discussed. For example, in the two examples, it would be useful to explain how Assumption 1 is satisfied with respect to problem-specific parameters like the arrival rate $\lambda_i$. Besides, how the desired prediction error can be achieved for specific prediction algorithms.

---

> ### Author Response · Authors · 2025-03-24
>
> We would like to thank the reviewer for taking the time to read the paper and provide some clarifications and discussions below.
>
> - The reviewer is right that we did not explicitly discuss the prediction for EV charging. In this example, prediction consists of two parts, energy price and arrival rate of EV, which are both region-dependent. A wide variant of studies has been conducted on the energy market of different regions. For instance, [McHugh et al., (2022)] studies the energy price prediction in Ireland, and  [Kapoor et al., (2025)] studies the energy price prediction in New Zealand. While few studies have explicitly studied the arrival rate of EV at charging stations, this problem shares a lot of common characteristics with the arrival rate of cars at parking garages, which is also commonly studied, see e.g. [Provoost et al., (2020)]. We have edited the relevant parts in the examples. We thank the reviewer for pointing out this gap.
> - The variance in the first example is to be expected. This example has very high randomness due to the setup of the arrival rate modeled by the Poisson process, which essentially models a 1-dimensional random walk. The arrival rate of jobs fluctuates in the experiment setup, which essentially models the change of arrival rate with time across the day. This also has to do with the low regret of the system: for a system with two servers and a constant job arrival rate, RSRT is proven to be the optimal policy, see e.g. [Özkan \& Kharoufeh, (2014)]. We examine the setting with three servers and a fluctuating job arrival rate, and it is to be expected that RSRT is still very close to the optimal policy.
> - The time horizons for the first and second examples are 100 and 50, respectively.  We have added clarification in the respective texts.
> - Thank you for pointing out the problem with $\alpha$. Indeed, $\alpha$ is used to define the contraction, and $\gamma$ is the exact value in our case. For clarification purposes, we have unified the notation to $\gamma$.
> - Thank you, and we apologize for the typo. The $s'$ in Definition 4 (now Definition 3) should be replaced with $\{S^{(i)}\}_i$.
>
> In addition to the above, we have incorporated changes of the related work, discussing the current bounds of RL for non-stationary MDPs. For further illustrative purposes, we also added a time-varying RL algorithm as a benchmark in our simulation. We hope the above have answered your questions, and please let us know if you have further questions or suggestions.

---

### Review · Reviewer_1ych · 2025-03-20

**Summary Of Contributions:**

This paper proposed a method for non-stationary MDPs with (inexact) prediction models of the transition kernel. The authors showed that even if the prediction model can be wrong (requiring sub-exponential), a sublunar regret is possible. In the error-free case, an exponential-decay bound is given.

**Audience:**

Yes

**Broader Impact Concerns:**

None.

**Claims And Evidence:**

Yes

**Requested Changes:**

1. I'd like to see more explanation on the techniques using variation budget in comparison with the method proposed. I am not sure I understand the concept very well. The assumption on prediction model having sub-exponential error seems to have strong correlation to that. More detailed comparison is needed.
2. The assumptions include sub-exponential error w.r.t time horizon. More detail on the applicability of such assumption should be added.
3. Please elaborate on the difference between MPC works such as Lin et al. 2021 and the method proposed, if there's any, besides the state space and transition model assumptions.
4. Definition 1 should be renamed to Assumption 1.
5. Writing in definition 4 should be improved.
6. The paper misses conclusion section and discussion section on the experiment results.
7. No references given for the compared baseline methods.

**Strengths And Weaknesses:**

Strengths:
1. The topic seems interesting on using a prediction model for the non-stationary MDPs.
2. Theoretical analysis is given for the proposed method with regret error bound.
3. Numerical experiments are conducted for practical problem setups.
Weaknesses:
1. The authors emphasize using a single trajectory for learning. But why can't the prior work that resets the controller be used on such scenarios as well? It seems naturally it forms a transfer learning setting where the environment keeps evolving, and the policy continuously use TD to update. Only focusing on a single trajectory seems to limit the scope of this work. Additionally, it does not make sense to me to consider a single trajectory setting with finite-horizon MDPs as each trajectory automatically resets.
2. It seems the proposed method is too idealistic, in the sense that we already assume having a predictive model which is close to the true transition kernel. From the perspective of model-based RL, I find it hard to distinguish its contribution as literature in model-based learning approaches already explored such methods extensively. What this paper is doing is even easier than model-based RL methods.
3. For the experiments, the authors did not compare with baselines RL methods.

---

> ### Author Response · Authors · 2025-03-24
>
> We would like to thank the reviewer for taking the time to read the paper. We believe that there might be some misunderstanding and would like to provide some clarifications below.
>
> - The setting of finding optimal policy on a single trajectory is relevant and has been studied in previous RL literatures e.g. [Wei \& Luo (2021); Jali et. al., (2025)] and control literatures [Lin et. al., (2021)]. Algorithms that work for multiple episodes/trajectories can not be easily adapted to the single trajectory setting, precisely because there is no reset of state at the end of each trajectories, since there is only one trajectory. Therefore, any mistake the agent makes during the learning process will be propagated to the end of the time horizon. In contrast, any mistake the agent makes in episodic RL only affects one episode/trajectory. Previous works in RL for this setting fix a variation budget in the form of $\sum_{t=1}^{T-1} \Vert P_t - P_{t-1}\Vert + \Vert r_t - r_{t-1}\Vert < \Delta_T$ for some fixed $\Delta_T$ [Wei \& Luo (2021); Jali et. al., (2025)], which is a typical assumption for time-varying RL on a single trajectory, whereas we do not enforce this assumption.
> - This paper is different from the model-based RL in that model-based RL methods can only learn from past information and thus always lag behind the current transitional probabilities and rewards, let alone take into account future system dynamics. To the best of the authors' knowledge, no previous result has shown any exponential decay of regret with respect to the prediction horizon. We further showed that even if the prediction is inaccurate, as long as the prediction error grows sub-exponentially with the prediction horizon, the regret will still be upper bounded. The closest work we find is [Merlis (2024)], which also uses prediction, but does not show any exponential decay property. We updated the related work section to discuss some most up to date bound in time-varying RL to show case the strength of our result.
> - The MPC works, such as [Lin et. al. (2021)], designs an algorithm that solves traditional control problems, which inherently has assumptions such as continuity of state space and a linear, deterministic transition kernel, whereas in the RL setting, we do not have the above assumptions. The above difference requires a fundamentally different analytical method, as you can see in the entirely different proof setups, such as the rule of span semi-norm in proving the exponential decay. Practically, it is clear that [Lin et. al. (2021)] would have a hard time solving the two examples proposed in this work, as both have discrete state spaces and nonlinear system dynamics. Moreover, MPC works, such as [Lin et. al. (2021)], generally have a cost function that requires the agent to stay around the origin, whereas in our problem, there is no assumption on the reward function, so the optimal states at each time step can be drastically different.
> - Thanks for your suggestion of comparing to time-varying RL algorithm. We added a comparison to a new time-varying RL algorithm in a single trajectory finite time horizon setting [Jali et. al., (2025)] and added relevant discussions. In the first submission, we have not compared with the time-varying RL works because it would not be fair to them. As mentioned in the review, the proposed algorithm enjoys significant advantage by having predictions of the future, which is not enjoyed by typical RL algorithms. To have a fair comparison, we are comparing to Fast Available Server (FAS) [Lin \& Kumar, (1984)] and ratio-of-service-rate-thresholds (RSRT) [Özkan \& Kharoufeh, (2014)], which are two of the most widely used algorithms in server allocation, with the latter proven to be optimal in two-servers case. Both of the above algorithms have full access to the current reward and transitional probability, which makes a fair comparison. Lastly, we note that the newly added time-varying RL algorithm's performance is very dependent on the initial policy, even after we pre-trained an initial policy close to RSRT to optimize its performance.
>
> We have also changed the related work to discuss more clearly the difference between the proposed algorithm and previous works in time-varying RL, explaining the difference in theoretical bounds achieved by our algorithms. We have also added a conclusion section. The discussion section is integrated into the simulation section, as it might be clearer to discuss the difference in performance immediately after introducing the settings. We hope the above has clarified some misunderstandings. Please do not hesitate to ask us if you have any more questions or suggestions.

---

### Review · Reviewer_ocKi · 2025-04-20

**Summary Of Contributions:**

The paper studies the look-ahead prediction for dynamic programming in non-stationary MDPs. By leveraging the look-ahead predictions of the model, the authors propose a MPC-style algorithm that takes the first action in a finite-horizon prediction. In theory, the authors prove that the regret bound of the proposed method that depends on the prediction error, time horizon, and the diameter of non-stationary MDPs. Finally, the authors provide an experiment to show improved regret by increasing prediction horizon.

**Audience:**

Yes

**Claims And Evidence:**

Yes

**Requested Changes:**

See detailed comments on Weaknesses: the notion of regret, technical challenge, and interpretation of the regret bound should be better explained.

**Strengths And Weaknesses:**

Strengths:

(1) The authors present an application of look-ahead prediction in dynamic programming. Theoretically, the authors prove that the look-ahead prediction can help with reducing regret bound.

(2) The authors show effectiveness of the proposed method in two applications.

Weaknesses:

(1) The regret analysis is misleading. The definition (2) is the optimality gap, not usual regret for online MDPs.

(2) The novelty in analysis should be highlighted, for instance how the contraction in span semi-norm works for look-ahead prediction.

(3) It is hard to interpret the regret bound in Theorem 4.1. It should be made more explicit when the regret bound is meaningful.

---

> ### Author Response · Authors · 2025-04-22
>
> We would like to thank the reviewer for taking the time to read the paper and would like to provide some clarifications below.
>
> - Our definition of regret is known as dynamic regret, which is widely used in time-varying RL, e.g. [Chandak et al., (2020)], which measures the performance difference between the proposed algorithm and the optimal dynamic policy, which is a stronger regret than typical static regret e.g. [Auer et al., (2008)].
>
> - Thanks for your suggestion to highlight the novelty of analytical methods. Indeed, using the contraction of the span semi-norm is the key to achieving the stated exponential decay in regret. We showed that the estimated value function is very close to the optimal value function in the span semi-norm. Therefore, we also showed that the estimated $Q$ function is very close to the optimal $Q$ function. Since the optimal policy uses $\arg\max$ on the optimal $Q$ function and the proposed policy uses $\arg\max$ on the estimated $Q$ function, we showed that those two policies are very similar. We are updating the discussion to make this novelty more explicit.
>
> - We apologize for the complexity of Theorem 4.1, which is a result of the dynamic prediction error in Assumption 1, which varies with the prediction time horizon. The key interpretation is that the regret decays exponentially with prediction time horizon, and a longer prediction time horizon always helps, given that the prediction error growth is sub-exponential. To make our result easier to understand, we append Corollary 4.2, which is our result in a noiseless setting. The regret bound in Corollary 4.2 demonstrates the exponential decay of regret, though it does not demonstrate the relationship between regret and prediction errors.
>
> We hope the above has answered your questions. Please let us know if you have further questions. In the next few days, we will upload an updated version of the paper incorporating the aforementioned changes.

---

### Decision · Action_Editor_rpMy · 2025-04-28

**Recommendation:** Accept with minor revision

**Comment:**

The reviewers are concerned about the restrictive assumptions, the lack of error analysis in the prediction step, and whether the proposed algorithm outperforms prior art in practice, not just theory, given the restrictive assumption. The work imposes a multi-step contraction, which facilitates the theory and is interesting. However, it is important to note the proposed algorithm has not been shown to outperform works for the stationary setting with off the shelf change detectors employed for restarting, or even methods for the stationary setting that use function approximators that are well-attuned to time-series (such as RNNs or non-stationary kernels in RKHS). Therefore, the I share the concerns of the reviewers and have determined the paper is not publishable as it is.

If the authors can substantially revise the paper to resolve all reviewer concerns, especially with respect to baselining the proposed approach, contrasting notions of convergence/regret with more standard usage in the literature, and demonstrate its performance on stationary and non-stationary settings in experiments, I would consider a revision of this work.

**Audience:**

yes

**Claims And Evidence:**

As mentioned in the below response field: not all claims are supported by sufficient evidence. In particular:

The authors have not demonstrated that the proposed algorithm outperform works for the stationary setting with off the shelf change detectors employed for restarting, or even methods for the stationary setting that use function approximators that are well-attuned to time-series (such as RNNs or non-stationary kernels in RKHS). Therefore, the I share the concerns of the reviewers and have determined the paper is not publishable as it is.

If the authors can substantially revise the paper to resolve all reviewer concerns, especially with respect to baselining the proposed approach, contrasting notions of convergence/regret with more standard usage in the literature, and demonstrate its performance on stationary and non-stationary settings in experiments, I would consider a revision of this work.

**Resubmission Of Major Revision:**

The authors may consider submitting a major revision at a later time.